# Effect of Solvent Polarity on the Spectral Characteristics of 5,10,15,20-Tetrakis(p-hydroxyphenyl)porphyrin

**DOI:** 10.3390/molecules28145516

**Published:** 2023-07-19

**Authors:** Hongwei Guo, Xianhu Liu, Lan Li, Yanping Chang, Wanqing Yao

**Affiliations:** School of Chemistry and Environment, Jiaying University, Meizhou 514015, China; guohw624@jyu.edu.cn (H.G.); liuxhv@126.com (X.L.); lanlee123@163.com (L.L.); changyp09@163.com (Y.C.)

**Keywords:** solvent effect, UV-vis absorption spectroscopy, vibrational spectroscopy, porphyrin, *E*_T_(30) parameters

## Abstract

The electronic absorption and vibrational spectra of deprotonated 5,10,15,20-tetrakis(p-hydroxyphenyl)porphyrin (THPP) are studied as a function of solvent polarity in H_2_O-DMF, H_2_O-acetone, H_2_O-methanol, and DMF-acetone mixtures. The maximum absorption wavelength (*λ*_max_) of the lowest energy electronic absorption band of deprotonated THPP shows an unusual solvatochromism-a bathochromic followed by a hypsochromic shift with reduced polarity. According to the correlation analysis, both specific interactions (H-bonds) and nonspecific interactions affect the spectral changes of this porphyrin. Furthermore, the solvent polarity scale *E*_T_(30) can explain both shifts very well. At higher polarity (*E*_T_(30) > 48), THPP exists as a hyperporphyrin. The *E*_T_(30) is linear with *λ*_max_ and a decrease in solvent polarity is accompanied by a bathochromic shift of *λ*_max_. These results can be rationalized in terms of the cooperative effects of H-bonds and nonspecific interactions on the spectra of hyperporphyrin. At relatively low polarity (45.5 < *E*_T_(30) < 48), hyperporphyrin gradually becomes Na_2_P as *E*_T_(30) reaches the critical value of 45.5. The spectrum of the hyperporphyrin turns into the three-band spectrum of the metalloporphyrin, which is accompanied by a hypsochromic shift of *λ*_max_.

## 1. Introduction

Porphyrins are a very important class of 18 π-electron-conjugated macrocycles that exhibit a variety of interesting optical, electrical, and physicochemical properties. Porphyrins are widely distributed in biological organisms, such as cytochromes, heme, and chlorophyll, and play an important role in physiological function. With their strong UV-vis absorption capacity, high quantum yield and singlet-oxygen-generating ability, good photostability, and high electron mobility, porphyrins are of increasing interest in the fields of photovoltaic cells and photodynamic therapy [1]. More importantly, porphyrins can form complexes with a variety of metal cations (M = Fe(II, III), Co(II, III), Mn(II, III, IV), etc.). Meanwhile, porphyrins and their metal derivatives have been widely used as the basic units of porous materials such as metal-organic frameworks (MOFs) and covalent organic frameworks (COFs). This makes porphyrins and their derivatives promising for a variety of applications, for example, as biosensors, catalysts, and for drug delivery and chemical storage [2,3,4,5]. 5,10,15,20-tetrakis(p-hydroxyphenyl)porphyrin (THPP) is a derivative of porphyrin that plays important roles in photosynthesis and electron transfer [6]. Furthermore, it can be modified at the meso-position to obtain targeting properties in biomedical applications involving photodynamic therapy (PDT), especially in cancer therapy [7,8,9,10]. Therefore, its photophysical and photochemical behavior continues to attract much interest.

It is well known that its spectrum varies with pH because it has two types of ionizable protons (Hs) [11], the comparatively acidic phenolic-Hs in the peripheral region and pyrrolic-Hs in two N-H groups. When the effect of pH on its spectrum in different solvents is studied, it is found that the solvent also has a profound effect on the spectrum of THPP. In a 50% DMF-50% H_2_O mixture [12], the deprotonation of THPP occurs only on the peripheral *p*-hydroxyphenyl groups, forming a hyperporphyrin. Martin Gouterman distinguishes three types of porphyrins: normal, hypso, and hyper. Hyperporphyrin spectra show prominent extra absorption bands in the region *λ* > 320 nm [2]. These extra features are not *π*-*π** transitions of the porphyrin ring, but due to charge transfer (CT) transitions such as from porphyrin to metal(d) or from porphyrin substituent to porphyrin or otherwise. In this deprotonated THPP, the lowest energy absorption band is attributed to the *n* (phenoxide anion)-*π** (porphyrin) charge transfer transition. When deprotonated in DMF [13], THPP can be further deprotonated from the pyrrolic-Hs and coordinates with two sodium ions to form the sodium complex of THPP (Na_2_P), converting the hyperporphyrin spectrum of THPP to the three-band spectrum of metalloporphyrin.

In fact, a comprehensive study of the dependence of the spectrum of the deprotonated THPP on solvent polarity has not been reported so far. In this work, the variation of the electronic and vibrational spectra of the deprotonated THPP with solvent polarity has been systematically investigated. Reichardt defined “solvent polarity” as “the overall solvation capability (or solvation power) for reactants and activated complexes as well as for molecules in the ground and excited states, which in turn depends on the action of all possible, specific and nonspecific, intermolecular forces between solvent and solute molecules, including Coulomb interactions between ions, directional interactions between dipoles, and inductive, dispersion, hydrogen-bonding, and charge transfer forces, as well as solvophobic interactions. Only those interactions leading to definite chemical alterations of the solute molecules through protonation, oxidation, reduction, complex formation, or other chemical processes are excluded” [14]. That is, the intermolecular interactions between the solvent and the solute molecules are roughly divided into nonspecific interactions (e.g., dipole-dipole, dipole-induced dipole interactions, and dispersion forces) and specific interactions (e.g., H-bonds in hydroxylic solvents) [14]. Therefore, four solvent pairs are chosen to cover several solute-solvent interactions: a proton acceptor and a hydroxylic solvent (DMF and water; acetone and water), two hydroxylic solvents (methanol and water), and two proton acceptors (DMF and acetone). Meanwhile, a solvent polarity scale *E*_T_(30), which is one of the more popular empirical scales [15,16], is used to characterize the polarity of these binary mixed solvents. We find that the maximum absorption wavelength (*λ*_max_) of the lowest-energy electronic absorption band of deprotonated THPP exhibits an unusual solvatochromism in these binary mixed solvents. To clarify the molecular origin of these changes, we consider two possible correlations: (i) *λ*_max_ and the C-O stretching frequency; (ii) *λ*_max_ and the empirical scale of the solvent polarity *E*_T_(30). In this deprotonated THPP, the n-electron is localized on the oxygen atoms of the hydroxyphenyl group [11], and oxygen atoms with n-electrons are prone to form hydrogen bonds with hydroxylic solvents such as water and methanol. The formation of hydrogen bonds (hydrogen-bond acceptor) is accompanied by a loss of electron density at the oxygen atom and therefore by a decrease in the frequency of the C-O stretching. In other words, the change in C-O stretching frequency is mainly caused by hydrogen bonding effects. *E*_T_(30) is based on the extremely solvatochromic character of 2,6-diphenyl-4-(2,4,6-triphenyl-1-pyridiniumyl)phenoxide (denoted in the literature as Reichardt’s Dye # 30) and is defined as the molar transition energy in kcal/mol of the intramolecular charge-transfer band of this dye in different solvents [17]. Kosower suggested that spectral shifts of strongly absorbing solutes (such as Reichardt’s Dye # 30 in this article) in various solvents might be used to establish a scale of solvent polarity [18]. It is generally accepted that the effect of a solvent on the spectrum of a solute is the resultant of a large number of factors-static factors such as interaction between solvent and solute permanent dipoles, dynamic factors such as dispersion forces, and specific interactions such as hydrogen bonding between solute and solvent [18]. As can be seen from Figure 1a, Reichardt’s Dye # 30 has a large dipole in the ground state with the n-electron localized on the phenolate oxygen atom and it is recognized that *E*_T_(30) is a descriptor of both hydrogen bond and nonspecific interactions between the solvent and Reichardt’s Dye # 30 [19]. We expect that the correlation analysis will clarify whether the solvatochromism is caused by specific effects or both, and we further discuss the mechanism of influence of the solvent on the spectral variation of THPP.

## 2. Results

### 2.1. Effect of Solvents on UV-Vis Spectra

Figure 2 shows the UV-vis spectral changes of THPP in four series of binary mixed solvents ([OH^−^] = 0.04 mol/L). The inset shows the dependence of the *λ*_max_ of the lowest energy band of THPP on the solvent composition.

Figure 2a shows (in H_2_O-DMF mixtures) that the spectral shape of THPP remains essentially the same when the volume percentage of DMF is less than 98%. As expected (inset), the lowest energy absorption band shows a solvatochromic phenomenon: *λ*_max_ gradually increases from 666 nm in water to 703 nm in 90% DMF. When the volume percentage of DMF is increased to 98%, the spectrum of THPP is completely transformed. That is, the hyperporphyrin spectrum becomes a three-band spectrum of metalloporphyrin [13], which has an opposite effect on *λ*_max_ (from 703 in 90% DMF to 673 nm in 98% DMF).

In H_2_O-acetone mixtures (Figure 2b), the spectral changes of THPP from H_2_O to acetone are similar to those reported above for the H_2_O-DMF mixtures. However, the solvent-dependent shift of the lowest energy absorption band is relatively small (inset: *λ*_max_ shifts from 666 nm in water to 685 nm in 90% acetone and then to 676 nm in 98% acetone). Note here that the shape of the spectrum of THPP in 98% acetone resembles a hyperporphyrin spectrum rather than a metalloporphyrin spectrum.

In H_2_O-methanol mixtures (Figure 2c), the spectral shape of THPP remains essentially unchanged from H_2_O to methanol and is attributed to hyperporphyrin spectra. The change in *λ*_max_ is rather small, shifting by only 3 nm (from 666 nm in water to 669 nm in 98% methanol).

In DMF-acetone mixtures (Figure 2d), the variation of the spectra is more complex: the spectral shape of THPP up to 20% acetone is similar to that in the 2% H_2_O-98% DMF mixture described above. In the range of 40–80% acetone, the spectrum is also similar to that in the 2% H_2_O-98% DMF mixture, but with a relative hypsochromic shift and a decrease in intensity. Further addition of acetone gives a spectrum similar to that in the 2% H_2_O-98% acetone mixture.

Interestingly, these spectral changes are reversible. Thus, depending on the composition of the starting solvent, the lowest energy absorption band of the deprotonated THPP may exhibit a bathochromic or hypsochromic shift, or both shifts.

### 2.2. Effect of Solvents on Vibrational Spectra

To clarify the effect of the solvent on the molecular structure, resonance Raman (RR) experiments were carried out. Figure 3 shows the RR spectra of THPP in the range of 900–1600 cm^−1^ in alkaline solutions ([OH^−^] = 0.04 mol/L) of (A) 100% H_2_O, (B) 80% DMF-20% H_2_O, (C) 80% acetone-20% H_2_O, (D) 80% methanol-20% H_2_O (E) 98% acetone-2% H_2_O, (F) 98% methanol-2% H_2_O, (G) 80% acetone-0% DMF, and (H) 98% acetone-2% DMF using a 514.5 nm excitation. The assignment of the RR spectra is based on the previous results and is shown in Table 1 [11,12,13]. The RR spectral data of THPP in the alkaline 98% DMF-2% H_2_O mixture in Table 1 are from Ref. [13].

In H_2_O-DMF, H_2_O-acetone and H_2_O-methanol mixtures, from H_2_O to 80% organic solvent (Table 1), the skeletal vibrations of THPP are almost unaffected by changes in solvent composition, except that the *ν*_9_ mode at 1084 cm^−1^ (mainly involving C_β_-H bending vibrations) shifts down by 11, 13, and 6 cm^−1^ to 1073, 1071, and 1078 cm^−1^, respectively [13]. The results of RR indicate that in highly aqueous solvents, the deprotonation of THPP occurs only on the peripheral *p*-hydroxyphenyl groups to form hyperporphyrin and the solvent has little effect on the structure of the central macrocycle of hyperporphyrin, implying that the H-bonding interaction of the solvent with the central N-H groups of hyperporphyrin is rather weak. The volume percentage of the organic solvent is further increased to 98%. In the 98% DMF-2% H_2_O mixture, the *ν*_19_ and *ν*_11_ modes (mainly involving C_α_-C_m_ and C_β_-C_β_ stretching vibrations) shift down by 43 and 17 cm^−1^ to 1509 and 1472 cm^−1^, respectively. The downshifted *ν*_19_ and *ν*_11_ modes suggest that THPP is further deprotonated of pyrrolic-Hs to form Na_2_P [13]. However, the *ν*_19_ and *ν*_11_ modes show no significant downshift in the 98% acetone-2% H_2_O and 98% methanol-2% H_2_O mixtures, suggesting that THPP cannot be converted to Na_2_P in either solution.

In DMF-acetone mixtures, the RR spectra of THPP in the range of 0–80% acetone are similar to those of metalloporphyrin (Na_2_P) in 98% DMF-2% H_2_O. As the volume percentage of acetone is further increased to 98%, the characteristic bands of Na_2_P disappear and the RR spectra are similar to those of hyperporphyrin in highly aqueous solutions.

To determine the effect of the solvent on the peripheral phenoxide anion substituents, we performed similar FTIR experiments on THPP. Unfortunately, in several nonaqueous solvents (98% DMF-2% H_2_O, 98% acetone-2% H_2_O, and DMF-acetone mixtures), the FTIR data could not be collected due to the low solubility of deprotonated THPP. Figure 4 shows a plot of the C-O stretching frequency of THPP versus the volume percentage of the organic solvents in three series of binary mixed solvents ([OH^−^] = 0.04 mol/L). It was expected that the intermolecular H-bonds would form between the C-O groups and the hydroxylic solvents, which is supported by the FTIR data. The formation of hydrogen bonds is accompanied by a loss of electron density at the oxygen atom and therefore by a decrease in the frequency of the C-O stretching. In H_2_O-DMF and H_2_O-acetone mixtures, the C-O stretching frequency gradually increases when increasing the volume percentage of the organic solvents. However, increasing the volume percentage of methanol in H_2_O-methanol mixtures leads to a decrease in C-O stretching frequency, despite the fact that H_2_O is a stronger H-bond donor than methanol. We attribute this phenomenon to the effect of self-association of H_2_O, which reduces the amount of free OH required for interaction with the dye [18,20]. The RR and FTIR data suggest that only the C-O groups of the peripheral substituents are affected by the H-bonding. The skeletal structure of hyperporphyrin is not greatly affected by the solvent, but the solvent polarity reaches a critical value and the skeletal structure changes significantly, converting the hyperporphyrin to Na_2_P.

## 3. Discussion

The existence of strong H-bonds between the C-O groups of the deprotonated THPP and the hydroxylic solvents, as demonstrated by RR and FTIR data, precludes the use of a model based on the Onsager theory of dielectrics [21]. Furthermore, when the H-bonding interaction is present, it usually dominates over nonspecific interactions [22,23]. Therefore, we focused our attention on two possible correlations of *λ*_max_: (i) the C-O stretching frequency; (ii) the empirical scale of the solvent polarity *E*_T_(30).

### 3.1. Correlation with the C-O Stretching Frequency

Due to incomplete FTIR data, this correlation cannot be used to interpret the spectral transformation of THPP in the less aqueous solvent mixtures. We expect that it can rationalize the solvatochromism of the lowest-energy electronic transition band (*n* (phenoxide anion)-*π** (porphyrin) CT transitions) of deprotonated THPP in highly aqueous solvent mixtures. A plot of the *λ*_max_ of the *n*-*π** transition band versus the corresponding C-O stretching frequency in three series of binary mixed solvents is shown in Figure 5. As can be seen from the plot, the decrease in C-O stretching frequency is accompanied by a hypsochromic shift of *λ*_max_ in H_2_O-DMF and *H_2_O*-acetone mixtures, but a bathochromic shift of *λ*_max_ occurs in H_2_O-methanol mixtures. Traditionally, the hypsochromic shift of the *n*-*π** absorption band is attributed to the formation of H-bonds from the hydroxylic solvent to the oxygen atoms of the phenoxide anions, and the formation of H-bonds is then thought to lower the energy of the n-orbital by an amount equal to the hypsochromic shift relative to a nonhydrogen bonding solvent [24,25]. As the volume percentage of H_2_O increases, the H-bonding interaction is stronger, the C-O stretching frequency is smaller, and *λ*_max_ is hypsochromically shifted. Therefore, the decrease in C-O stretching frequency is accompanied by a hypsochromic shift of *λ*_max_.

It is thereby rather unexpected that a reversal trend is observed in H_2_O-methanol mixtures. These observations seem to suggest that the above explanation is overly simplistic and that other effects may be superimposed on and dominate the H-bonding effect. We consider these other effects to be mainly nonspecific interactions and previous studies have shown that the n-electrons are localized on the peripheral C-O groups in the ground state, and the charges are delocalized into the macrocycle upon excitation [11]. In addition, the peripheral C-O groups are not in the same plane as the macrocycle, so this hyperporphyrin has a larger dipole moment in the ground state. As a result, the ground state of the hyperporphyrin is more solvated than the excited state by the solute-solvent nonspecific interactions (dipole-dipole and dipole-induced dipole interactions) in polar media. Therefore, the *λ*_max_ of the *n*-*π** transition band is hypsochromically shifted with the increasing solvent polarity [26,27]. As the volume percentage of H_2_O decreases in the H_2_O-methanol mixtures, the C-O stretching frequency is smaller, but *λ*_max_ is bathochromically shifted due to the decrease in solvent polarity. Therefore, the decrease in C-O stretching frequency is accompanied by a bathochromic shift of *λ*_max_. According to the correlation analysis between *λ*_max_ and the C-O stretching frequency, both specific interactions (H-bonding) and nonspecific interactions affect the spectral changes of this hyperporphyrin, and the spectral phenomena cannot be well explained by considering only H-bonding interactions.

### 3.2. Correlation with the E_T_(30) Scale

From these considerations, we searched for an alternative parameter that could better explain and predict the effects of solvents on the spectra of deprotonated THPP. One such solvent parameter considered was the *E*_T_(30) scale. The strongly basic character of this dye suggests that the parameter *E*_T_(30) includes specific solvent acidity effects as well as nonspecific effects [17]. Moreover, deprotonated THPP also has a large dipole and a strongly basic character in the ground state. It appears that this dye-solvent interaction is similar to the deprotonated THPP-solvent interaction. *E*_T_(30) is therefore a more appropriate measure of the interactions between deprotonated THPP and solvent molecules.

Figure 6 shows the relationship between the *λ*_max_ of THPP in four series of alkaline binary mixed solvents and the *E*_T_(30) values of the solvent mixtures. In the case where *E*_T_(30) > 48 (THPP exists as hyperporphyrin), *λ*_max_ correlates well with *E*_T_(30) according to Equations (1)–(3) shown below:
*λ*_cal_ = −2.63 *E*_T_(30) + 829.94 
         *r* = 0.999 *S*_n_ = 0.57 n = 10   H_2_O − DMF
(1)

*λ*_cal_ = −1.37 *E*_T_(30) + 753.33 
          *r* = 0.997 *S*_n_ = 0.46 n = 10   H_2_O − acetone
(2)

*λ*_cal_ = −0.42 *E*_T_(30) + 692.68 
           *r* = 0.970 *S*_n_ = 0.30 n = 11   H_2_O − methanol
(3)


This result is expected as it is known that *E*_T_(30) can account for H-bonding interactions as well as nonspecific interactions. However, unlike the correlation of *λ*_max_ with the C-O stretching frequency, the decrease in solvent polarity from H_2_O to organic solvent is accompanied by a bathochromic shift of *λ*_max_ in three series of mixtures, and this difference can be explained as follows. According to the above discussion, both H-bonding and nonspecific interactions affect the spectral changes of this hyperporphyrin. Therefore, the effects of these two factors on the spectral changes are discussed separately. (1) Nonspecific interactions: The foregoing results suggest that the nonspecific interactions cause a bathochromic shift of *λ*_max_ with a decreasing solvent polarity. It can be seen from Figure 6 that the polarity of the solvent decreases from H_2_O to organic solvents in three series of mixtures. (2) H-bonding effect: Similarly, we already know that the H-bonding effect leads to a bathochromic shift of *λ*_max_ as the H-bonding effect weakens. As can be seen in Figure 4, the H-bonding effect from H_2_O to the organic solvents is weakened in the *H_2_O*-acetone and H_2_O-DMF mixtures, while it is strengthened in the H_2_O-methanol mixtures. Thus, in both the H_2_O-acetone and H_2_O-DMF mixtures, nonspecific interactions and the H-bonding effect lead to a bathochromic shift of *λ*_max_ from H_2_O to the organic solvent, which is the observed direction. In the H_2_O-methanol mixture, these two effects are reversed, and the nonspecific interactions are relatively larger, resulting in a relatively smaller bathochromic shift of *λ*_max_.

For low-polarity solvents (*E*_T_(30) < 48), in H_2_O-DMF and H_2_O-acetone mixtures, *λ*_max_ deviates increasingly from Eqs. (1) and (2) as *E*_T_(30) decreases. In H_2_O-DMF mixtures, this deviation is due to the formation of Na_2_P [13]. In H_2_O-acetone mixtures, however, THPP does not become Na_2_P (Figure 3E). There are two possible explanations for this discrepancy: (1) the solvent mixtures have different chemical compositions; (2) the solvent mixtures have different *E*_T_(30) values. As shown in Figure 2d and Figure 3G, in the DMF-acetone mixture, the hyperporphyrin becomes Na_2_P once the volume percentage of acetone is less than 80%. On the other hand, Figure 6 shows that the *E*_T_(30) value decreases slightly from acetone to DMF. Based on the above observation, (2) should be responsible for the formation of Na_2_P. It seems to us that *λ*_max_ deviates more and more from the equation as *E*_T_(30) approaches the critical value (from 48 to 45.5). Once *E*_T_(30) exceeds the critical value, a completely structural change from hyperporphyrin to Na_2_P occurs. This is probably due to the fact that THPP has two types of ionizable protons (H), the comparatively acidic phenolic-H in the peripheral region and pyrrolic-H in the two N-H groups. The N-H groups are very weakly acidic (pK > 15) [13]. This acid-base reaction, in which THPP is deprotonated by NaOH, is in equilibrium between a soft acid (porphyrin) and a hard base (NaOH), so solvent effects can alter their relative strengths. Hard bases are strongly stabilized by intermolecular interactions between the solvent and the solute molecules (especially H-bonds in hydroxylic solvents), whereas soft acids are not much affected by intermolecular interactions, and RR experiments have demonstrated that the central N-H groups of hyperporphyrin are not greatly affected by the solvent [13]. Decreasing the polarity of the solvent, for example from water to DMF, increases the base strength of the NaOH but hardly affect the acid strength of the porphyrin. Thus, once *E*_T_(30) exceeds the critical value, THPP can be further deprotonated from the pyrrolic-Hs, thereby destroying the hyperporphyrin effect. Here, the spectrum of Na_2_P is the normal metalloporphyrin spectrum for which the solvent effect should be quite complicated. As shown in Figure 2d, in the range of 100–20% DMF in the DMF-acetone mixture (THPP exists as Na_2_P), the variation of the spectrum is quite complex and no useful correlation between *E*_T_(30) and *λ*_max_ can be obtained. This fact is likely attributed to the formation of different ionic aggregates in the less aqueous solvent mixtures [28,29].

These results suggest that the *E*_T_(30) scale can predict the effect of the solvent on the spectra of deprotonated THPP. At higher polarity (*E*_T_(30) > 48), THPP exists as a hyperporphyrin. *λ*_max_ is linearly correlated with *E*_T_(30) and a decrease in solvent polarity is accompanied by a bathochromic shift of *λ*_max_. These results can be rationalized in terms of the cooperative effects of H-bonds and nonspecific interactions on the spectra of hyperporphyrin. At relatively low polarity (45.5 < *E*_T_(30) < 48), hyperporphyrin gradually becomes Na_2_P as *E*_T_(30) reaches the critical value of 45.5. In solvents of lower polarity (*E*_T_(30) < 45.5), THPP exists as Na_2_P. A poor correlation is obtained between *E*_T_(30) and *λ*_max_. This is attributed to the formation of different ionic aggregations in the less aqueous solvent mixtures.

In this work, the unusual solvatochromism of deprotonated THPP in solvents of different polarity has been studied, and the molecular origin of the spectral change has been investigated by RR and IR. However, the mechanism of the complex spectral changes of Na_2_P in low-polarity (*E*_T_(30) < 45.5) solvents remains unclear and requires further investigation, while computational chemistry can better support the interpretation of the results. The hyperporphyrin systems based on deprotonated THPP exhibit the attractive spectra for solar absorption, and the results can be used to guide the design of novel porphyrin photosensitizers for use in photovoltaic cells and photodynamic therapy.

## 4. Experimental Section

### 4.1. Materials

THPP was prepared as previously described [11]. The labeling of specific carbon atoms on the macrocycle is shown in Figure 1b.

Reichardt’s Dye # 30 was purchased from Aldrich Chemical Co. (used as received). Analytical grade N, N-dimethylformamide (DMF), acetone, and methanol were vacuum-distilled before use. Double-distilled water was used for all sample preparations. All other chemicals used were analytical grade and were used as received.

### 4.2. Raman and FTIR Spectra

Raman spectra were recorded at room temperature using a Renishaw RM2000 Raman spectrophotometer. Binary mixed solvents were prepared in a volumetric ratio. Samples containing 5 × 10^−4^ mol/L THPP and 0.04 mol/L NaOH were placed in the capillary tube. Radiation of 514.5 nm (incident power of 20–50 mW) was focused on the detection area from the side of the capillary tube. Spectra were collected from 900 to 1600 cm^−1^ with a 15 s exposure time, applying a resolution of about 1 cm^−1^. The resulting RR spectrum was obtained by solvent subtraction. FTIR spectra were obtained using a Bruker VERTEX 70v FTIR spectrometer (Germany). The concentrations of THPP and NaOH in three series of mixed solvents were 5 × 10^−3^ mol/L and 0.04 mol/L, respectively. Samples were analyzed using an AquaSpec flow cell with calcium fluoride (CaF_2_) windows, a 0.6 mm spacer, with a resolution of 4 cm^−1^ at room temperature. For each sample, 128 consecutive scans collected from 1650 to 1100 cm^−1^ were averaged to achieve the final spectrum. The resulting FTIR spectrum was obtained by subtracting the spectral background given by the solvent and NaOH. Each sample was scanned three times.

### 4.3. UV-Vis Spectra

UV-vis absorption spectra were taken on an Agilent Cary 300 UV-vis spectrophotometer. UV-vis spectra of THPP and Reichardt’s Dye #30 were obtained using a 1 cm quartz cell. The concentrations of THPP and NaOH in four series of mixed solvents were 4 × 10^−6^ mol/L and 0.04 mol/L, respectively. The concentration of Reichardt’s Dye #30 in each system was 0.10 mmol/L except for pure water, which was saturated with the dye. Each sample was scanned four times. The position of the maximum of the first absorption band for Reichardt’s Dye #30 was measured to ±l nm and converted to *E*_T_(30) using Equation (4)
*E*_T_ (30) (kcal·mol^−1^) = 28,591/(*λ*_max_/nm)(4)

## Figures and Tables

**Figure 1 molecules-28-05516-f001:**
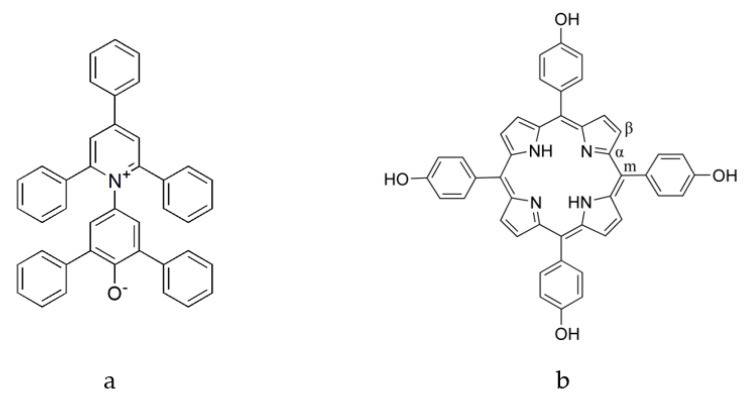
Structure of (**a**) Reichardt’s Dye # 30 and (**b**) THPP.

**Figure 2 molecules-28-05516-f002:**
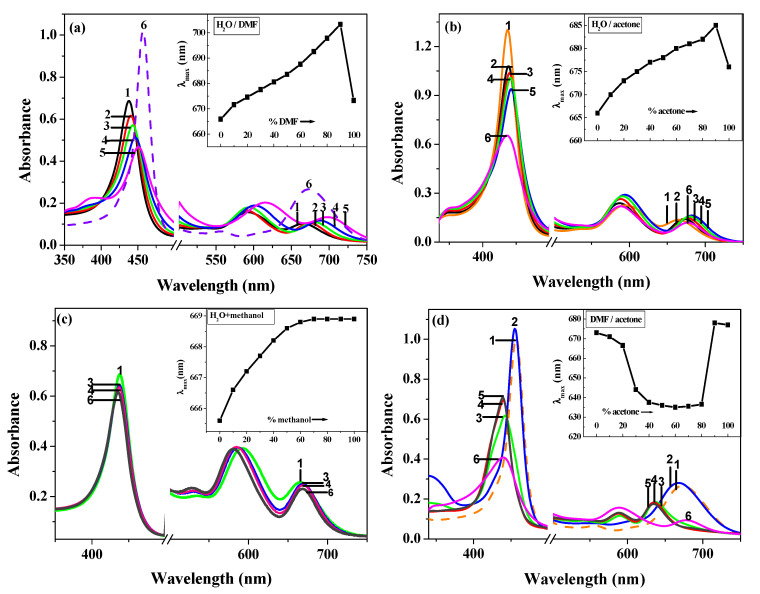
Spectral characteristics of THPP in the Soret band and Q-band region at [OH^−^] = 0.04 mol/L as a function of the volume percentage of *X* expressed in (**a**) *X*_DMF_, (**b**) *X*_acetone_, (**c**) *X*_methanol_, and (**d**) *X*_acetone_: (1) 0, (2) 20, (3) 40, (4) 60, (5) 80, (6) 98. Inset: different variations of *λ*_max_ of THPP with the volume percentage of *X*.

**Figure 3 molecules-28-05516-f003:**
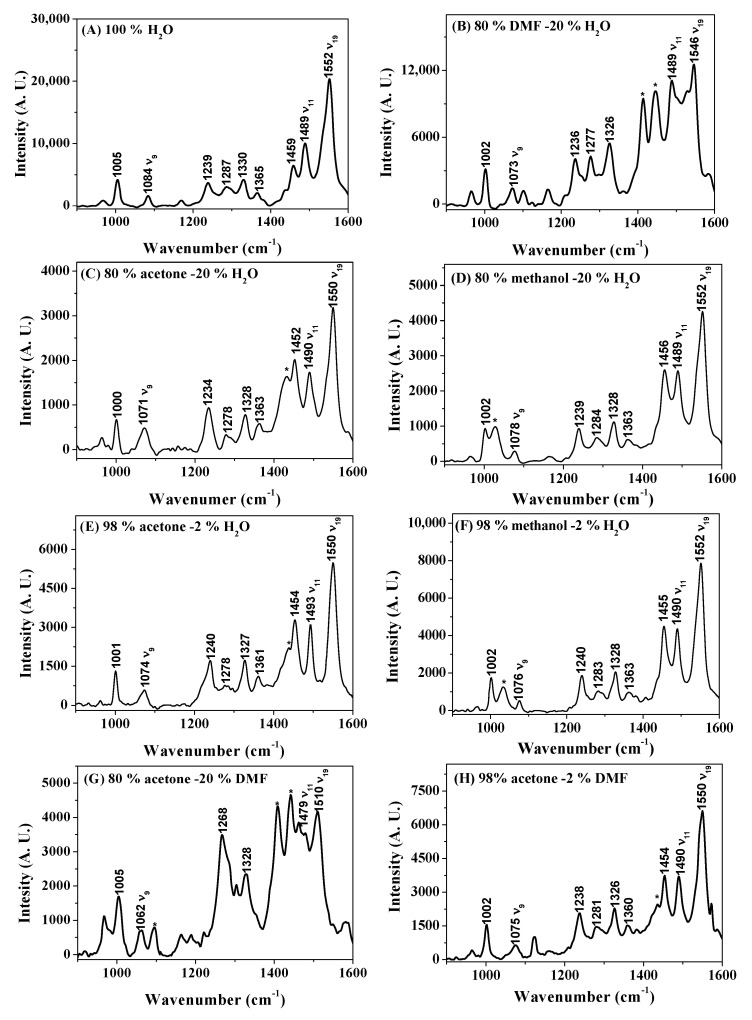
Resonance Raman spectra of THPP in the range of 900–1600 cm^−1^ in alkaline solutions ([OH^−^] = 0.04 mol/L) of (**A**) 100% H_2_O, (**B**) 80% DMF-20% H_2_O, (**C**) 80% acetone-20% H_2_O, (**D**) 80% methanol-20% H_2_O, (**E**) 98% acetone-2% H_2_O, (**F**) 98% methanol-2% H_2_O, (**G**) 80% acetone-20% DMF, and (**H**) 98% acetone-2% DMF using a 514.5 nm excitation. The small negative peaks are due to solvent subtraction or noise. * = solvent band.

**Figure 4 molecules-28-05516-f004:**
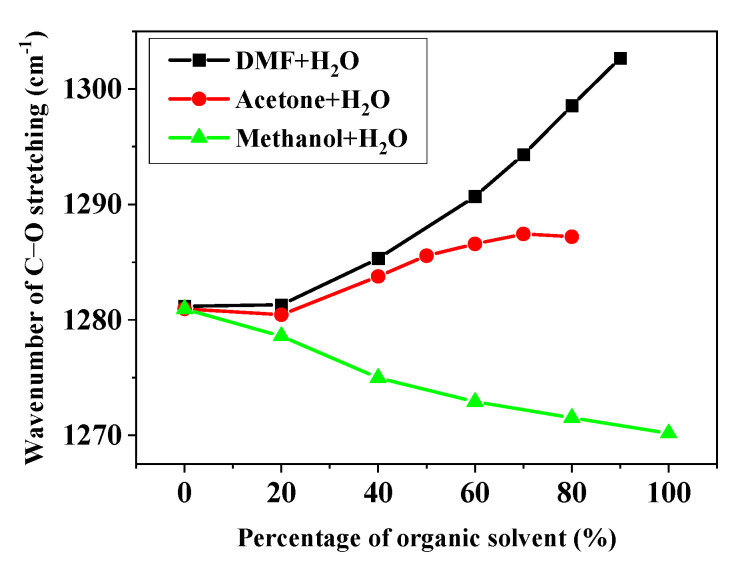
Relationship between the C-O stretching frequencies of the phenoxide anion substituents and the volume percentage of the organic solvents in three series of the mixed solvents.

**Figure 5 molecules-28-05516-f005:**
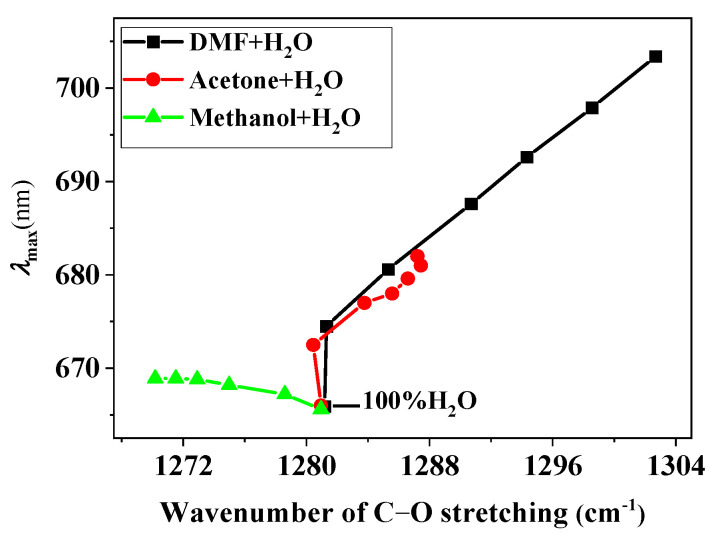
Relationship between the *λ*_max_ of the *n*-*π** transition band and the corresponding C-O stretching frequency of the phenoxide anion substituents in three series of mixed solvents.

**Figure 6 molecules-28-05516-f006:**
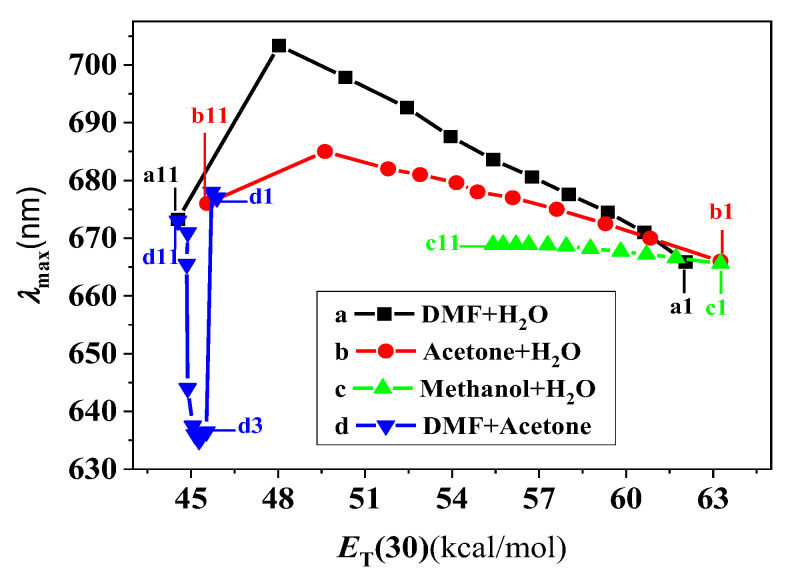
Relationship between *λ*_max_ and *E*_T_(30) value of the solvent in four series of the mixed solvents. The solvent composition at each point of the four curves is expressed as the volume percentage of *X* (a) *X*_DMF_, (b) *X*_acetone_, (c) *X*_methanol_, and (d) *X*_DMF_: (1) 0, (2) 10, (3) 20, (4) 30, (5) 40, (6) 50, (7) 60, (8) 70, (9) 80, (10) 90, (11) 98.

**Table 1 molecules-28-05516-t001:** Wavenumber of Raman shift (cm^−1^) of THPP in alkaline solutions.

A	B	C	D	E	F	G	H	I ^a^	Assignment
1552	1546	1550	1552	1550	1552	1510	1550	1509	*ν*(C_α_C_m_) + *δ*(C_α_C_m_C_Ph_)(*ν*_19_)
1489	1489	1490	1489	1493	1490	1479	1490	1472	*ν*(C_β_C_β_) + *δ*(C_β_H) (ν_11_)
1330	1326	1328	1328	1327	1328	1328	1326	1327	*ν*(C_α_C_β_) + *δ*(C_β_H)
1239	1236	1234	1239	1240	1240		1238		*ν*(NC_α_) + *ν*(C_α_C_β_) + *δ*(C_α_C_β_C_β_) + *δ*(C_α_C_m_)
1084	1073	1071	1078	1074	1076	1062	1075	1061	*δ*(C_β_H) + *ν*(C_β_C_β_) (ν_9_)
1005	1002	1000	1002	1001	1002	1005	1002	1003	*ν*(C_α_C_β_) + *ν*(NC_α_) + *ν*(CC)_ph_

^a^ Data for THPP in 98% DMF-2% H_2_O are from Ref. [13].

## Data Availability

Data is contained within the article.

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
