# Peer review of "Effect of Solvent Polarity on the Spectral Characteristics of 5,10,15,20-Tetrakis(p-hydroxyphenyl)porphyrin"

_molecules, 2023, doi:10.3390/molecules28145516_

Round 1
Reviewer 1 Report
Review of manuscript
molecules-2469790-peer-review-v1
The manuscript under review, titled "Effect of Solvent Polarity on the Spectral Characteristics of 5, 10, 15, 20-Tetrakis(4-hydroxyphenyl)porphyrin", by Hongwei Guo, Xianhu Liu, Lan Li, Yanping Chang and Wanqing Yao, describes an investigation of the solvent effect on the vibrational electronic spectra of the title compound.
Generally, the article is mostly well-structured and some interesting conclusions are derived. It is my opinion that the work could be be publishable in "Molecules" after a major revision, addressing the following questions:
1.
My main objection is the confusion with the concept of polarity and its relation to ET(30): it is questionable that the ET(30) parameter includes specific interactions. The meaning of the "polarity" concept is misrepresented on the introduction. Is is being incorrectly regarded as encompassing all solvent effects.
Quoting from ref. 9, "The term “polarity” is usually related to the capacity of a solvent for solvating dissolved charged or dipolar species." Therefore the sentence "In general, solvent polarity refers to the solvation capability of a solvent." is a misquotation. The original sentence refers only to the solvation of charged and dipolar species, not to all solutes in general.
Ref. 10 is also misquoted. It just says that "the solvent effect can be divided into specific and nonspecific solute−solvent interactions", and that "The word polarity is used in this paper as referring only to nonspecific solute−solvent interactions." It does not say that "solvent polarity (...) encompasses all of the intermolecular interactions between solvent and solute molecules and is roughly divided into nonspecific interactions (e.g., dipole-dipole, dipole-induced dipole interactions, and dispersion forces) and specific interactions (e.g., H-bonds in hydroxylic solvents)".
Please see also the following reference:
- José P. Cerón-Carrasco, Denis Jacquemin, Christian Laurence, Aurélien Planchat, Christian Reichardt, Khadija Sraïdi, "Solvent polarity scales: determination of new ET(30) values for 84 organic solvents" Journal of Physical Organic Chemistry, 27(6) 2014, 512-518
https://doi.org/10.1002/poc.3293
I believe it could be useful in this discussion and should be cited appropriately.
2.
The chosen acronym for the title compound is easily mistaken for a chemical formula. I recommend an alternative, like for example "THPP", used in the cited literature.
3.
Revise the text processing. Pay attention to the use of spaces, capitalization, superscript/subscript uses. There are several instances of all these, which I will abstain to point out in detail.
Also important is the use of accepted rules of physics and chemistry: italic symbols for quantities, rules of chemical nomenclature, units in charts/plots, etc.
P.e. the units are missing in the vertical axes of figures 5 and 6 (nm) and in the horizontal axis of figure 6 (kcal/mol). In the title of table 1 and in figures 4 and 5, "wavenumber" should be used instead of "frequency", consistently with the figure 3 horizontal axes.
4.
The spectroscopy experimental conditions, namely for Raman and FT-IR spectra, should be much more detailed. There in no mention of resolution, scan accumulation, sampling methodology, concentrations, solvent spectra subtraction, etc.
5.
Please explain why the low solubility in several non-aqueous solvents (98% DMF-2% H2O, 98% acetone-2 % H2O and DMF-acetone mixtures) is a problem for the IR measurements and not for UV-vis.
6.
The designation of the direction of frequency/wavelength shifts in the spectra should be consistent. Preferably use the more generally rigorous bathochromic/hypsochromic instead of red/blue shift, which only really applies to visible spectrum.
7.
Also, the interpretation of the results could be much better supported with the use of computational chemistry calculations.
Pay attention to the use of spaces, capitalization, superscript/subscript uses. There are several instances of all these, which I will abstain to point out in detail.
There also typos throughout the text (p.e. "a bule shift" in the abstract)
Correct typesetting should be taken care by the authors and not by the referees.
Reviewer 2 Report
This manuscript presents an interesting study on the electronic absorption and vibrational spectra of deprotonated 5, 10, 15, 20-tetrakis(p- 7 hydroxyphenyl)porphyrin ((OH)4PH2) as a function of solvent polarity in H2O-DMF, H2O-acetone, H2O-methanol, and DMF-acetone mixtures. The authors have found that the maximum absorption wavelength (λmax) of the lowest energy electronic absorption band of (OH)4PH2 shows an unusual solvatochromism- a bathochromic followed by a hypsochromic shift with reduced polarity. The correlation analysis has revealed that both specific interactions (H-bonds) and nonspecific interactions affect the spectral changes of this porphyrin. Furthermore, the authors have shown that the solvent polarity scale ET(30) can explain both shifts well. At higher polarity (ET(30) > 48), (OH)4PH2 exists as a hyperporphyrin and at lower polarity (45.5 <ET(30) < 48), hyperporphyrin gradually becomes Na2P as ET(30) reaches the critical value of 45.5. The spectrum of the hyperporphyrin turns into the three-banded spectrum of the metalloporphyrin, which is accompanied by a blue shift of λmax. These results provide useful insights into the combined effects of H-bonds and nonspecific interactions on the spectra of hyperporphyrin. Overall, the authors have presented a well-structured and convincing study on the electronic absorption and vibrational spectra of (OH)4PH2 as a function of solvent polarity. -The manuscript can be improved by providing more detailed descriptions of the correlation analysis and solvent polarity scale ET(30) used in the study. -Additionally, further discussion on why the critical value of ET(30) is 45.5 would be beneficial. Is it the specific nature of compounds? -In the introduction, it is necessary to discuss recent works on porphyrins [ 10.3390/molecules28030964 10.1016/j.ica.2023.121638 10.3390/polym15041055] to emphasize the study's relevance for metal complexes of similar compounds. Finally, the authors should provide some conclusions on the implications of their results and suggest further research directions.
Reviewer 3 Report
The article of Hongwei Guo et al. describes the study of the spectral properties of deprotonated 5, 10, 15, 20-Tetrakis(4-hydroxyphenyl)porphyrin in a series of mixed solvents with different solute-solvent interactions to elicit the role of specific and unspecific interactions on the molecular structure of the dye. The investigation was carried out with the help of absorption spectroscopy and vibrational spectroscopy, and the influence of the solvent polarity was correlated using the ET(30) scale. The article is well-written and evolves some new peculiar properties in the spectral properties of the porphyrins, which are one of the most multifaceted and promising classes of organic compounds. Overall, I have no remarks related to the descriptions of experimental results or explanations of the obtained properties. A small note about the sequence of the presentation. In particular, in the Abstract and in the Introduction, there is no mention of the application of correlations of λmax based on the C-O stretching frequency, but it is presented in the Discussion (section 4.1). Probably, it can be mentioned in the Introduction section. Also a slight disadvantage is the modesty of the description of the merits and significance of porphyrins, while references can be made to reviews, for example, 10.1021/jacsau.2c00255. In total, the article will be of interest to researchers working with porphyrins and specialists in the field of analytical chemistry. I think that it can be published after a minor revision.
Round 2
Reviewer 1 Report
I am satisfied with the author's effort to improve the manuscript.
I just want to call their attention for a couple of minor details:
- In Figure 1a, R should be defined as phenyl.
- In figure 6, the "k" for "kilo" should not be capital. It sould be "kcal/mol", not "Kcal/mol"
Author Response
"Please see the attachment."
